# Diversity of Neurotransmitter-Producing Human Skin Commensals

**DOI:** 10.3390/ijms252212345

**Published:** 2024-11-17

**Authors:** Samane Rahmdel, Moushumi Purkayastha, Mulugeta Nega, Elisa Liberini, Ningna Li, Arif Luqman, Holger Brüggemann, Friedrich Götz

**Affiliations:** 1Microbial Genetics, Interfaculty Institute of Microbiology and Infection Medicine Tübingen (IMIT), University of Tübingen, 72076 Tübingen, Germanyelisa@liberini.de (E.L.);; 2Department of Biology, Institut Teknologi Sepuluh Nopember, Surabaya 60111, Indonesia; arif.luqman@its.ac.id; 3Department of Biomedicine, Aarhus University, 8000 Aarhus Centrum, Denmark; brueggemann@biomed.au.dk

**Keywords:** aromatic amino acid decarboxylase, microbiota, neurotransmitter, skin, trace amines

## Abstract

Recent findings indicate that human microbiota can excrete trace amines, dopamine, and serotonin. These neurotransmitters (NTs) can either affect classical neurotransmitter signaling or directly trigger trace amine-associated receptors (TAARs), with still unclear consequences for host physiology. Compared to gut microbiota, less information is available on the role of skin microbiota in NT production. To explore this, 1909 skin isolates, mainly from the genera *Staphylococcus*, *Bacillus*, and *Corynebacterium*, were tested for NT production. Only 6.7% of the isolates were capable of producing NTs, all of which belonged to the *Staphylococcus* genus. Based on substrate specificity, we identified two distinct profiles among the NT producers. One group primarily produced tryptamine (TRY) and phenylethylamine (PEA), while the other mainly produced tyramine (TYM) and dopamine (Dopa). These differing production profiles could be attributed to the activity of two distinct aromatic amino acid decarboxylase enzymes, SadA and TDC, responsible for generating the TRY/PEA and TYM/Dopa product spectra, respectively. SadA and TDC orthologues differ in structure and size; SadA has approximately 475 amino acids, whereas the TDC type consists of about 620 amino acids. The genomic localization of the respective genes also varies: *tdc* genes are typically found in small, conserved gene clusters, while *sadA* genes are not. The heterologous expression of *sadA* and *tdc* in *Escherichia coli* yielded the same product spectrum as the parent strains. The possible effects of skin microbiota-derived NTs on neuroreceptor signaling in the human host remain to be investigated.

## 1. Introduction

Monoamines including catecholamines (dopamine, epinephrine, norepinephrine), serotonin, melatonin, histamine, and trace amines (TAs) are a group of mammalian neuroactive compounds produced by both host cells and microbiota. TAs are neuromodulators of classical neurotransmitters (NTs), including dopamine, noradrenalin, and serotonin [1]. However, the recent discovery of TA-associated receptors (TAARs) suggests that TAs may also function as NTs in vertebrates [2,3]. While TAs activate these receptors with nanomolar affinities, classical NTs require micromolar concentrations for TAAR activation [4]. From this point forward, for simplicity, we will refer to all mentioned monoamines as NTs.

The main enzymes responsible for the production of these NTs are categorized as aromatic L-amino acid decarboxylases (AADCs) [1]. The bacterial AADCs differ from human ones with respect to substrate specificity and kinetics of activity [5,6]. Different kinds of microbial AADCs have been reported so far. Tyrosine decarboxylase (TDC) with the ability to decarboxylate tyrosine, dihydroxy phenylalanine (L-Dopa), and/or phenylalanine is the most common one among enterococci and other lactic acid bacteria [7,8,9,10,11,12,13,14,15]. *Ruminococcus gnavus*, an anaerobic Gram-positive gut bacterium from the *Clostridiales* family, utilizes a tryptophan decarboxylase to primarily produce tryptamine (TRY), but also phenylethylamine (PEA), and tyramine (TYM). The same enzyme has also been reported in another gut species, *Clostridium sporogenes* [6]. The AADC enzyme from *Bacillus atrophaeus* shows broad substrate activity, with tryptophan at 610%, tyrosine at 12%, L-Dopa at 24%, and 5-hydroxytryptophan (5-HTP) at 71%, all relative to its activity towards phenylalanine [16]. The AADC enzyme from *Pseudomonas putida* is exclusively active towards L-Dopa [17]. A highly unspecific AADC, SadA, has been described in certain *Staphylococcus* species belonging to the skin and gut microbiota. This enzyme is able to produce all three TAs (TRY, PEA, and TYM), as well as serotonin (SER), and dopamine (Dopa). However, the production of NTs varies significantly among *Staphylococcus* species, with some strains, such as *S. pseudintermedius*, exhibiting higher levels of production, while others produce considerably lower amounts [18,19]. A metagenomic analysis revealed the presence of SadA homologs in at least 7 phyla such as *Bacillota*, *Actinobacteria*, *Proteobacteria*, *Bacteroidetes*, and *Acidobacteria*, and in 23 genera of the phylum *Bacillota*, including species within the skin and the gut microbiota such as *Clostridiaceae*, *Enterococcaceae*, *Lactobacillaceae*, or *Ruminococcaceae* [19].

Despite the high levels of monoamines in stool and skin samples from healthy human subjects [20,21], it is difficult to determine the extent to which microbiota contribute to the NT content of the body. However, compelling evidence from mouse studies suggests that the gut microbiota significantly contributes to the NT content in the brain, blood, gut, and other tissues [22,23,24,25].

The advantages of monoamine production by bacteria are not fully understood; however, the AADC activity seems to be an acid resistance mechanism by consuming excessive cytoplasmic protons and producing alkaline products to control intracellular pH homeostasis [7,10,26,27,28]. Monoamines have also been shown to increase bacterial adherence and internalization into epithelial cells, thereby providing a colonization advantage [18,20,29]. Moreover, these metabolites may act as signaling molecules to interact with the host niche [30]. The effects of TRY on gut motility [31]; the anti-inflammatory effects of TRY and its metabolite, indole acetic acid, on macrophages and hepatocytes [32]; and the immunomodulatory effects of TYM on enterocytes [29] are examples of monoamine-mediated interactions between microbiota and the host. In addition, NT-producing *Staphylococcus epidermidis*, which belongs to the normal skin microbiota, was found to accelerate wound healing [21]. Notably, a recent study showed that *S. epidermidis* isolates from healthy skin differ from those originating from atopic skin, with these differences linked to their ability to produce agonists of aryl hydrocarbon receptors (AHRs). A further analysis confirmed the absence of genes encoding NT-producing enzymes (AADCs) in atopic skin isolates [33]. Moreover, microbial tryptophan metabolites, including tryptamine, may be able to suppress acne vulgaris-associated inflammation by regulating the innate immune response [34]. Microbiota-derived NTs are also assumed to affect the mental state and may be associated with neurodegenerative diseases [4,35,36].

Monoamines produced by human microbiota can, therefore, enter the bloodstream and exert their effects beyond the original niche. However, compared to gut microbiota, less attention has been given to the role of skin and nasal microbiota in NT production. Given the ability of TAs to passively diffuse through cell membranes [2] and the widespread distribution of TAARs throughout the body [37], the skin microbiota may significantly contribute to the microbiota-derived fraction of NTs in the body.

In this study, we aimed to investigate the spectrum of NT production by the skin microbiota and identify the enzymes involved in this process.

## 2. Results

Strain collection of bacteria from human skin: Swab samples were taken from the antecubital fossa skin of 30 healthy volunteers to isolate aerobic and anaerobic bacterial species on an agar medium as detailed in the Materials and Methods. The colony forming unit (**CFU**) counts varied tremendously across samples and ranged from 2.28 to 6.14 log CFU/100 cm^2^ skin area (Figure 1 and Appendix A). No statistically significant differences were found between samples derived from men and women with respect to the number of CFUs (*p* = 0.151). There was no significant correlation between age and CFU counts (r = −0.185, *p* = 0.329). Following the isolation and purification of colonies with different morphologies, a diverse strain collection of 1909 skin bacteria was established. No lactic acid bacteria (LAB) were isolated from MRS and M17 plates incubated under anaerobic conditions. Based on 16S rRNA sequencing, the spectrum of predominantly Gram (+) bacteria mainly comprised the genera *Staphylococcus*, *Bacillus*, and *Corynebacterium*, with the vast majority of the isolates belonging to the genus *Staphylococcus* (Table 1).

Identification of neurotransmitter (NT) producers: The identification of NT producers was performed in two steps. In a first screening, all 1909 isolates were cultivated in TSB overnight, and the resulting culture supernatants were then analyzed for the ability to convert aromatic amino acids (AAs) to their decarboxylation products by RP-HPLC [18]. From previous work, we knew that TSB has a relatively high content of tryptophan, phenylalanine, and tyrosine (367, 613, and 355 μg/mL, respectively), which serve as substrates of AADC enzymes [18]. This analysis enabled us to identify NT producers—mainly staphylococcal strains—in our isolate collection as shown in Table 2. Most staphylococcal NT producers belonged to the species *S. epidermidis*, followed by *Staphylococcus saccharolyticus*, *Staphylococcus coagulans*, *Staphylococcus capitis*, *Staphylococcus hominis*, and *Staphylococcus pragensis*.

To reduce the noise of the media components in RP-HPLC analyses, a different method for sample preparation was implemented. We washed the overnight cultures twice with 1 × PBS and then resuspended the pellets in 5 × PBS/10% glucose, together with all respective substrates (1 mg/mL each), L-Dopa, tryptophan (W), phenylalanine (F), and tyrosine (Y). Representative RP-HPLC analyses are shown for an *S. epidermidis* isolate, which mainly produces PEA and TRY and little TYM and Dopa (Figure 2A), and an *S. capitis* isolate, which mainly produces Dopa and TYM and little PEA (Figure 2B). With this refined production process, we could clearly distinguish between two classes of NT producers: class I members predominantly produced TRY and PEA and little TYM and Dopa, whereas class II members predominantly produced TYM and Dopa and little PEA (Figure 2C). Class I made up the largest group with 118 isolates, while class II comprised only 10 isolates. Of the 1909 skin isolates, 128 (6.7%) were able to produce TAs (Appendix A).

SadA and TDC enzymes are responsible for NT production: The class I production pattern had already been reported in staphylococcal isolates from the gut and skin and is known to be mediated by the enzyme SadA [18]. It was, therefore, reasonable to hypothesize that the same enzyme is responsible for NT production in the isolates investigated in this study. To test this hypothesis, we confirmed the presence of the *sadA* gene in the isolates with a class I production pattern by PCR using *sadA*-specific primer sets (Appendix A). The observed class II production pattern resembled that previously reported for another AADC enzyme type, TDC, that is common among food-related lactic acid bacteria, especially *Enterococcus* sp. [7,8,9,10,11,12,13,14,15]. The presence of the *tdc* gene in skin isolates exhibiting a class II production pattern was confirmed using a consensus primer set with high specificity for various Gram (+) bacteria, yielding the expected amplicon size of 335 bps [38].

SadA homologues are more prevalent in *Staphylococcus* spp: Following a BLASTp search of the NCBI NR protein sequence database within the *Staphylococcus* genus, using a threshold of 90% coverage and 50% identity, SadA and TDC homologues were found in 6098 staphylococcal strains (Table 3). Among these, 5900 strains had SadA homologues, 176 contained TDC homologues, and 22 strains carried both (Appendix A).

Identification of tdc operon in *S. epidermidis* skin isolate: Among the NT-producing isolates harboring the *tdc* gene, *S. epidermidis* 102 was selected for further characterization. The analysis of the whole genome sequence of this isolate revealed the presence of a 6409 bp *tdc* operon consisting of four contiguous open reading frames (ORFs) oriented in the same direction, but with a slightly different gene arrangement compared to previously reported *tdc* operons (Figure 3). The BLASTp analysis showed high amino acid sequence homology with the previously described *tdc* operon from *Levilactobacillus brevis* IOEB (Table 4).

The same *tdc* operon was also found in other TDC-harboring staphylococcal species retrieved from the NCBI database, except in *S. caprae* strains. The *tdc* operon of *S. caprea* strains consists of only two genes, *tdc* and *nhaC* (Figure 3). Based on the phylogenetic tree, all TDC enzymes were closely related but exhibited a substantial genetic distance from the SadA enzyme (Figure 4A). Multiple alignments of the aa sequences indicated high similarity between the TDC enzyme from *S. epidermidis* 102 and those from *Enterococcus durans* IPLA 655, *E. faecalis* JH2-2, *E. mundtii* QU 25, and *L. brevis* IOEB strains (Appendix A). In addition, the 3D structure models of these TDC proteins revealed a comparable folding pattern, which clearly differs from the SadA protein type (Figure 4B). The alignment of aa sequences and the 3D structures of TDC and SadA enzymes from *S. epidermidis* species showed relatively low similarity, with a root mean square deviation (RMSD) value of 4.410 Å (Figure 4C and Appendix A). A further difference between SadA and TDC orthologues is their size: SadA enzymes have an average monomer size of approximately 475 aa, whereas TDC enzymes are larger with 616–635 aa.

In *S. epidermidis* 102, downstream of *tdc* are three other ORFs encoding tyrosine–tyramine permease (*tyrP*), the Na^+^/H^+^ antiporter (*nhaC*), and tyrosyl-tRNA synthetase (*tyrS*) (Figure 3). Putative promoters and Rho-independent terminators were found upstream of the start codons of *tdc*, *nhaC*, and *tyrS*, but not between *tdc* and *tyrP*. This suggests that *tdc* and *tyrP* might be co-transcribed independently of *tyrS* and *nhaC*. The *nhaC* gene, which is generally found as a flanking gene upstream of *tyrP* [7,39], is located between *tyrP* and *tyrS* in the *tdc* operon of *S. epidermidis* 102. However, some *tdc* operons such as those from *E. faecalis* JH2-2 [40], *E. mundtii* [9], and *L. lactis* IPLA 655 [8] lack this gene. TyrS is a class I aminoacyl tRNA synthetase characterized by the conserved HIGH and KMSKS motifs. In TyrS of *S. epidermidis* 102, the KMSKS motif is replaced by KFGKT as in TryS of *E. faecalis*, *E. mundtii*, *L. brevis*, and *L. lactis* [8,9,11,40].

TDC is responsible for NT production in *S. epidermidis* 102: To confirm the role of TDC in NT production in *S. epidermidis* 102, we generated the deletion mutant 102Δ*tdc* and compared its NT production to that of the WT strain. After the overnight incubation of the WT and mutant cells with individual or a combination of substrates (W, F, Y, L-Dopa, and 5-HTP), the culture supernatants were collected and analyzed by HPLC. No NT production activity was observed following the deletion of the *tdc* gene. In contrast, the WT strain exhibited decarboxylase activity primarily towards Y and L-dopa, but also towards F. It was, however, unable to decarboxylate 5-HTP and W to SER and TRY. In the absence of Y and L-dopa, there was a higher accumulation of PEA. The production of TRY and SER was negligible even when W and 5-HTP were the only available substrates (Figure 5A). The same production pattern was found for the TDC-harboring *E. faecalis* strain (ATCC 19433), although the production of Dopa and PEA was lower due to the slower growth rate of *E. faecalis* compared to *S. epidermidis* (Appendix A).

The *tdc* genes from *S. epidermidis* 102 and *E. faecalis* strains were then cloned (Figure 6A) and overexpressed in *E. coli*. The resulting transformants were fed with individual substrates (W, F, Y, L-Dopa, and 5-HTP). No production was found in the *E. coli* strain with the empty vector, whereas the *E. coli* strains expressing either of the *tdc* genes showed AADC activity comparable to the original strains (Figure 5B and Appendix A).

In *S. epidermidis* 102, the *tdc* gene encodes a protein of 616 aa with an estimated molecular mass of 69.9 kDa, which was confirmed by a single band at an approximate molecular weight of 70 kDa in an SDS-PAGE analysis of the purified enzyme (Figure 6B). The TDC enzyme from *E. faecalis* ATCC 19433 consisting of 620 amino acids with a molecular weight of 70 kDa was cloned in *E. coli* and used as a control. The decarboxylase activity of the purified enzymes was time-dependent. The total conversion of Y (2 mM) to TYM occurred within less than 30 min, while the accumulation of Dopa and PEA was steady and gradual over 20 h. The respective production of TRY and SER by both enzymes was negligible or undetectable (Figure 6C).

## 3. Discussion

Metagenomic profiling of the human skin microbiome revealed that SadA homologs are widely distributed throughout almost the entire bacterial kingdom [19]. In this study, skin swabs were collected from the antecubital fossa of 30 volunteers and the isolated bacterial strains were examined for their ability to produce NTs. We created a strain collection of 1909 pure cultures, ensuring the inclusion of a diverse range of colony morphologies, although the applied culturing method did not capture the full bacterial diversity observed in metagenomic studies [41,42,43,44,45]. One possible explanation for the discrepancy between metagenomic and cultivation estimates of bacterial abundance on the skin [46,47] is that traditional metagenomic approaches and 16S rRNA sequencing do not differentiate between DNA from viable and non-viable bacteria, leading to an overestimation of both the richness and diversity of the skin microbiome [48].

Out of the 1909 skin isolates, only 6.7% were identified as NT producers. All NT-producing isolates belonged to the genus *Staphylococcus*, with *S. epidermidis* being the most prevalent species, which can be attributed to its high abundance on the skin compared to other staphylococcal species [43,44,49]. Previous studies have shown that not all *Staphylococcus* species produce NTs. Of the 44 species representatives tested, only 12 (27%) produced NTs [18]. Additionally, current evidence indicates considerable diversity in NT-producing abilities among different strains of the same species. Therefore, the absence of NT producers in bacterial genera other than *Staphylococcus* does not necessarily indicate that they lack strains harboring AADC enzymes. Indeed, AADC activity has been reported in taxonomically distant species available in our strain collection, including *Bacillus cereus* [50], *B. atrophaeus* [16], and *Streptococcus thermophilus* [14].

NT-producing staphylococci were categorized into two production classes: class I, which produces mainly TRY and PEA and only little TYM and DOPA, and class II, which produces mainly TYM and Dopa and only little PEA (Figure 2A,B). The class I product pattern was found to be the result of SadA activity, while class II was attributed to the activity of TDC.

The class II type (TDC enzyme) is underrepresented in our skin isolates; only 10 (7.8%) of the 128 NT-producing staphylococcal skin isolates showed this type (Figure 2C, Appendix A). We found that the TDC type is also underrepresented in the published staphylococcal genomes with a share of only 3.2% (Appendix A). The dataset only partially represents the staphylococci found on human skin, as many species such as *S. pseudintermedius*, *S. delphinii*, and *S. aureus* do not inhabit this environment. If these three species are excluded, the dominance of SadA becomes significantly less pronounced: with 1031 SadA-encoding strains compared to 191 TDC-encoding strains, the latter account for around 16% of the retrieved strains. Our results suggest that TRY and PEA are likely to be dominant on the human skin. However, in a previous study, TYM was found to be very abundant on the human skin [21]. This discrepancy suggests that, with the applied growth conditions, we probably missed species carrying the *tdc*-type gene, an aspect that will be investigated in future studies. On the other hand, in contrast to TYM and Dopa, the main products of TDC enzymes, the products of the SadA-type enzyme (TRY and PEA) can cross the blood–brain barrier (BBB) [51]. Microbiota-derived PEA has been shown to accumulate systemically, cross the BBB, and trigger lethal PEA poisoning in germ-free mice colonized with a PEA-producing strain, *Morganella morganii* [22]. Since *S. aureus* infections are also known to cause a variety of neurological complications, most of which affect the central nervous system (CNS) [52], the question is to what extent TRY and PEA contribute to these effects. Further studies are needed to clarify the role of skin microbiota-derived TRY and PEA in CNS-related complications.

Although all AADCs are PLP-dependent homodimeric enzymes [35], SadA and TDC orthologues differ in size, phylogenetic relationship, and predicted 3D structure. While the SadA-type monomer is ~475 aa in length, the TDC type comprises ~620 aa. The phylogenetic tree of TDC enzymes from various staphylococcal and non-staphylococcal species shows that these proteins are closely related and distinct from the SadA cluster (Figure 4A). The separation of SadA orthologues from TDC enzymes in the phylogenetic tree is also reflected in the AlphaFold 2 structure predictions (Figure 4B).

The genomic localization of *sadA* was analyzed in 15 staphylococcal species [18]. There was neither a common insertion site, nor was the gene flanked by mobile elements, indicating a lack of horizontal transfer of *sadA*. In contrast, *tdc* is flanked by genes such as *nhaC*, *tyrP*, or *tyrS*, that are also present in unrelated bacteria, including different staphylococcal species, *E. durans*, *E. faecalis*, *E. faecium*, *E. hirae*, *E. mundtii*, and *L. brevis* (Figure 3 and Table 4) [7,8,9,10,11,13], suggesting that the *tdc* gene cluster has spread by horizontal gene transfer. Whether these flanking genes form a functional unit is not entirely understood. Besides their essential catalytic roles in protein biosynthesis, aminoacyl-tRNA synthetases like TyrS also contribute to other functions, such as the regulation of gene expression [8]. For example, in enteric bacteria, the expression of histidine biosynthetic genes is regulated by histidyl-tRNA synthetase as a sensor for the intracellular histidine pool [53]. The *tyrS* gene in the *tdc* operon was proposed to regulate decarboxylation in response to tyrosine concentration and extracellular pH [8,54]. Like TYM-producing *E. faecalis* V583 and *L. brevis* ATCC 357 [8], *S. epidermidis* 102 has a related TyrS paralog with 54% identity to the TyrS in the *tdc* operon, supporting the regulatory role of the protein encoded in the operon. The role of NhaC in the biosynthesis of TYM has yet to be determined [55]. However, as part of the *tdc* operon, it might also be involved in maintaining cytoplasmic pH homeostasis by uptaking Na+ in exchange for proton extrusion [28]. The permease TyrP catalyzes electrogenic exchange of tyrosine and TYM [56]. In all bacteria with a *tyrP*-containing *tdc* operon, *tdc* and *tyrP* are co-transcribed. It is assumed that amino acid decarboxylation coupled with amino acid/amine antiport generates proton motive force at the cell membrane [56,57]. The generation of metabolic energy due to the combined action of histidine decarboxylase and the histidine/histamine antiporter was reported in *Lactobacillus buchneri,* suggesting an indirect proton pumping mechanism as the means of producing metabolic energy in bacteria [58]. Although we expressed only the *tdc* gene without *tyrP* from *S. epidermidis* 102 in *E. coli*, there was no difference in the product spectrum. Since *E. coli* BL21 has a tyrosine transporter (TyrP, GenBank accession no. ACT43729.1), the role of TyrP as a necessary TYM transporter remains unclear. TYM, PEA, and TRY were shown to cross synthetic lipid bilayer membranes by simple diffusion [59]. TYM can pass through the intestinal epithelial cells via passive diffusion but cannot cross the BBB, whereas PEA is able to cross the BBB [2]. Further studies are, therefore, required to address the possible role of transporters in the release of TAs. This will help determine whether variations in NT production spectra among bacteria are due to differences in AADC enzyme specificity or the available transport systems.

The specificity of the SadA enzyme has already been studied, and as previously mentioned, the main products of SadA are TRY and PEA [18]. In this study, the TDC enzymes of *S. epidermidis* 102 and *E. faecalis* ATCC 19433 were purified from *E. coli* and tested for their substrate spectrum. Both enzymes showed the same substrate specificity: the highest conversion rate was observed with the substrate Y, followed by F, and then L-Dopa. Hardly any turnover was detected with W and 5-HTP, the precursor of serotonin. Bacterial TDC enzymes show different NT production spectra; while all show high affinity for Y, their affinities for other substrates vary. TDC orthologues from *L. brevis* CGMCC 1.2028, *E. faecalis*, and *E. faecium* show 2–10-fold higher catalytic efficiency towards Y compared to L-Dopa [5,15,60], whereas the TDC from *L. brevis* IOEB 9809 cannot decarboxylate L-Dopa [61]. Enterococcal TDCs can also produce PEA, albeit with a very low conversion rate and yield [57,62]. The TDC enzymes from *L. brevis* strains (CGMCC 1.2028 and IOEB 9809) are, however, unable to produce PEA [15,61].

Comparison between *S. epidermidis* 102 and its Δ*tdc* mutant indicates that its TDC (Figure 5A), similar to that of *E. faecalis*, is capable of decarboxylating Y, L-Dopa, and phenylalanine. However, unexpectedly, the purified enzymes from both *S. epidermidis* 102 and *E. faecalis* exhibited lower activity towards L-Dopa compared to the parent strains or the corresponding *E. coli* transformants. In this study, the enzymatic assays were conducted at pH 6.8, while the maximum activity of TDC enzymes occurs at lower pH levels. The purified TDC enzyme from *Lactobacillus brevis* CGMCC 1.2028 (GenBank accession no. AFP73381.1) was shown to have the highest activity at pH 5.0, with activity decreasing at pH levels above 6.0 [15]. A similar optimum pH range has been reported for TDC enzymes from *E. faecalis*, *E. faecium*, and *L. brevis* IOEB 9809 [5,60,61]. Since AADCs are cytoplasmic enzymes, their activity should be assessed at intracellular pH (pH_i_) levels. In LABs, the intracellular pH can drop to as low as 5 in response to a decrease in extracellular pH [28,63,64]. In contrast, staphylococci are able to maintain a more neutral pH_i_ in an acidic environment [65,66,67]. However, to thrive in acidic niches like the skin (pH 4.2 to 5.9) [67], staphylococci may possess mechanisms that enhance AADC activity, as increasing decarboxylation activity is a bacterial strategy to overcome environmental acid stress [7,10,26,27,28]. Moreover, in our experiments, the bacterial cells were incubated with 10% glucose. Glucose fermentation acidifies the cytoplasm, which can then induce the tyrosine decarboxylation process to stabilize the internal pH [28].

This study was limited by a relatively low sample size (30 individuals), the collection of samples from a single skin site, and the use of culture-dependent assays. Furthermore, the cultivation conditions used in this study may not have been optimal for detecting certain bacterial species potentially involved in NT production. A combination of culture-based and metagenomic analyses of microbial communities from different skin sites could provide a comprehensive view of the skin microbiota and NT-producing commensals. Additionally, the whole genome sequencing of skin commensals might yield valuable insights into the evolutionary dynamics of AADCs within the skin microbiota. Although the presence of NT-producing bacteria was identified, further research is required to investigate the potential effects of skin microbiota-derived NTs on host neurological signaling pathways, which may contribute to a range of physiological and psychological functions, including the immune response, inflammation, skin health, mental state, and pathogenesis of neurodegenerative diseases.

## 4. Materials and Methods

Materials: All media and chemicals were of either an analytical or ultrapure grade from Merck Co. (Darmstadt, Germany) unless otherwise stated. Q5 High-Fidelity DNA polymerase and NEBuilder HiFi DNA Assembly Master Mix were from New England Biolabs Inc. (Beverly, MA, USA). Restriction enzymes were provided by Thermo Fisher Scientific (Waltham, MA, USA). Glass beads were purchased from Carl Roth GmbH + Co. (Karlsruhe, Germany).

Skin swab sample collection and cultivation: The skin swab samples were collected from 30 healthy volunteers aged between 22 and 76 years (15 Females and 15 Males). None of the subjects had a history of skin disease or had received any treatment within at least 3 months before this study. A pre-moistened cotton swab was used to sample a 100 cm^2^ area of the antecubital fossa. Swabs were then vortexed in PBS (pH 7.2) and diluted up to 10,000-fold and the bacterial suspensions were inoculated onto agar media. Tryptic soy broth agar (TSBA, TSB plus 15 g L^−1^ agar) plates were used to isolate a wide variety of aerobic species and incubated for 2 days at 30 °C under aerobic conditions (Heraeus Instruments, Hanau, Germany). To isolate LABs, the samples were inoculated onto De Man–Rogosa–Sharpe (MRS) and M17 plates and incubated at 37 °C under anaerobic conditions for 3 days [68]. Furazolidone–Tween 80–Oil red O (FTO) agar plates, incubated at 37 °C under an aerobic atmosphere with 5% CO_2_ (Heraeus Instruments, Hanau, Germany), were used to isolate corynebacteria [69]. Colonies with different morphologies from TSA, MRS, and M17 plates, along with orange-pink circular colonies from FTO plates, were subcultured into pure cultures using the same agar plates and under the same incubation conditions. The bacterial count in the sampled area was determined as log CFU/100 cm^2^ using TSBA cultures.

Identification of NT producers: For primary screening, colonies from TSBA plates were inoculated into TSB and incubated at 37 °C under aerobic conditions for 2 days with shaking at 150 rpm. Corynebacteria isolates were cultivated in a brain–heart infusion broth containing 1% Tween 80 under an aerobic atmosphere with 5% CO_2_ at 37 °C for 2 days. The supernatants were collected every 24 h and analyzed using reversed-phase HPLC (RP-HPLC, Agilent Technologies, Waldbronn, Germany) at room temperature with a Poroshell 120 EC-C18 HPLC column (Agilent; 4.6 × 150 mm, 2.7 µm). Linear gradient elution was performed at a flow rate of 1 mL/min, transitioning from 100% of 0.1% phosphoric acid to 100% of 50% acetonitrile over 15 min, with an injection volume of 10 µL. The diode array detector was set at 210 nm, with 360 nm used as the reference [21]. Cell pellets from overnight cultures of producing isolates were washed with PBS (pH 7.2) and resuspended in 5 × PBS (pH 7.2) supplemented with 10% glucose and 1 mg/mL of substrates (W, F, Y, and L-Dopa) at an OD of 50 [18]. After overnight incubation under the same conditions, the supernatants were subjected to the RP-HPLC analysis. *S. epidermidis O47* and *E. faecalis* ATCC 19433 were used as positive controls. The NT peaks were confirmed by spiking the samples with NT’s standards and comparing retention times and UV spectra.

The producing isolates were then identified at the species level by amplifying a 1525 bp fragment of the 16S rRNA gene using the universal primers E8F (5′-AGAGTTTGATCCTGGCTCAG-3′) and E1541R (5′-AAGGAGGTGATCCANCCRCA-3′) [70]. Amplification was conducted in a 50 µL reaction mixture containing 10 µL of 5× Q5 reaction buffer, 0.2 mM dNTPs, 0.5 µM of each primer, 1 unit of Q5 DNA polymerase, and 50 to 100 ng of a DNA template. The thermal cycling program consisted of an initial denaturation step of 30 s at 98 °C, followed by 35 cycles of 10 s at 98 °C, 30 s at 50 °C, and 90 s at 72 °C, with a final elongation at 72 °C for 2 min. PCR products were sequenced at both ends at Eurofins Genomics (Ebersberg, Germany). The 16S rRNA sequences were then aligned against the EzTaxon database (http://www.ezbiocloud.net/eztaxon) (accessed on 23 August 2023) for species identification.

To gain an overview of the spectrum of bacteria isolated from the skin, a selection of non-producing isolates with varying morphologies was also identified.

Identification of *sadA* and *tdc* genes in NT producers: To detect the *sadA* gene in NT-producing isolates, three *sadA*-specific primer sets were designed by aligning sequences from the NCBI database using Geneious Prime 2019.1.3 (Biomatters Limited, Auckland, New Zealand) (Appendix A).

DEC3 (5′-CCGCCAGCAGAATATGGAAYRTANCCCAT-3′) and DEC5 (5′-CGTTGTTGGTGTTGTTGGCACNACNGARGARG-3′) primers were used for the detection of the *tdc* gene [38].

The PCR conditions were identical to those used for the 16S rRNA PCR except that the elongation step was set to 24 s.

Construction of *tdc* deletion mutant: The temperature-sensitive shuttle vector pBASE6 was used to construct the *S. epidermidis* 102 mutant lacking the *tdc* gene (102Δ*tdc*) [18]. The 1 kb flanking regions of the *tdc* gene were amplified from *S. epidermidis* 102 genomic DNA with primer pairs KOS1F/KOS1R and KOS2F/KOS2R (Appendix A). The linearized plasmid (*EcoRV* restriction site) was then ligated with the fragments using Hi-Fi DNA Assembly Master Mix. The constructed plasmid was first transformed into *E. coli* DC10B and then into *S. epidermidis* 102 by electroporation. Positive transformants were identified by colony PCR using the KOS3F/KOS3R primer pair. Mutagenesis was carried out by the method of Bae and Schneewind [71] and mutants were confirmed by a sequence analysis. The NT production of the 102Δ*tdc* strain was investigated as mentioned above.

Overexpression and purification of TDC: The *tdc* genes from *S. epidermidis* 102 and *E. faecalis* ATCC 19433 were amplified with primer pairs 78S/79S and 32S/33S, respectively (Appendix A). The amplicons were then ligated into a *BmtI* and *NotI* digested pET28a using Hi-Fi DNA Assembly Master Mix. The pET28a plasmids encoding N-terminal His-tagged *tdc* genes were first transformed into *E. coli* DC10B and then into *E. coli* BL21 expression strains. The transformants were confirmed by colony PCR using primer sets 81S/82S and 42S/44S and sequencing.

For expression, overnight cultures of recombinant BL21 strains were diluted in a fresh LB medium supplemented with kanamycin (30 µg/mL) to an OD_600_ of 0.1. The cultures were grown to the exponential phase (OD_600_ of 0.6) and induced with 0.2 mM Isopropyl-*β*-D-1-thiogalactopyranoside (IPTG) at 15 °C for 16–18 h. The cells were then collected, resuspended in a lysis buffer (300 mM NaCl, 10 mM imidazole, 50 mM KPO_4_, protease inhibitor mix; pH 7.5), mixed with a mixture of 0.1 and 1 mm glass beads (1:1), and lysed using a Precellys^®^ Evolution homogenizer (Bertin Technologies, Montigny-le-Bretonneux, France). The lysate was cleared by centrifugation at 40,000× *g* at 4 °C for 1 h. The 6 × His-tagged proteins were purified using a nickel-nitrilotriacetic acid (Ni-NTA) agarose matrix (Qiagen, Hilden, Germany) as described by van Kessel et al. [72]. Protein concentrations were measured using a NanoPhotometer^®^ NP80 spectrophotometer (Implen, Munich, Germany), with the predicted extinction coefficient and molecular weight from the ExPASy ProtParam tool (https://web.expasy.org/protparam/). The purified proteins were confirmed by a 12% SDS-polyacrylamide gel electrophoresis (SDS-PAGE) analysis.

In vitro enzymatic assay: The catalytic activity of the purified enzymes towards W, F, Y, L-DOPA, or 5-HTP was determined in a reaction mixture of 100 µL containing 50 mM sodium phosphate (pH 6.8), 300 mM NaCl, 40 µM PLP, 150 nM enzyme, and 2 mM substrate. The reactions were quenched at different time intervals with methanol or ethyl acetate and analyzed by RP-HPLC [18].

In silico analyses: A BLASTp search of the NCBI NR protein sequence database was conducted on 15 May 2023 to identify homologues of the SadA enzymes from *S. epidermidis* O47 (GenBank accession no. QKN61770.1) and *S. schleiferi* NCTC 12218 (GenBank accession no. CAD7359406.1), as well as the TDC enzyme from *E. faecalis* JH2-2 (GenBank accession no. AAM46082), within the genus *Staphylococcus*. Alignments with a minimum identity of 50% over at least 90% of the protein length were considered putative homologues.

Genome sequencing of *S. epidermidis* 102 was conducted using Illumina shotgun sequencing at the Göttingen Genomics Laboratory. DNA libraries were prepared using the Nextera XT DNA sample preparation kit and then sequenced on a MiSeq platform (Illumina, San Diego, CA, USA) utilizing the v3 reagent kit with 600 cycles. Raw reads were trimmed using Trimmomatic version 0.39. The assembly of reads into contigs was performed using SPAdes genome assembler software (version 3.15.2). The coverage and quality of the assembled genome were estimated using Qualimap version 2.2.1. The draft genome sequence was deposited in GenBank under the accession number JBIENV000000000. Gene prediction and annotation were performed with RAST [73].

Sequence similarity searches and alignments were performed using BLASTp and Clustal O (version 1.2.4), respectively. The web tool ARNold was used to predict the presence of Rho-independent terminators in *tdc* operons (http://rssf.i2bc.paris-saclay.fr/toolbox/arnold/) (accessed on 11 July 2024). Intergenic regions were also analyzed for the presence of δ^54^ and δ^70^ promoters with iPro54-PseKNC (http://lin-group.cn/server/iPro54-PseKNC) (accessed on 11 July 2024) and iPro70-PseZNC (http://lin-group.cn/server/iPro70-PseZNC) (accessed on 11 July 2024).

A maximum likelihood phylogenetic tree of AADC enzymes (SadA and TDC) was constructed using the MEGA_11.0.13 package with 1000 bootstrap replicates. The 3D structures of the enzymes were predicted using AlphaFold 2 and subsequently aligned using the PyMOL alignment command (version 3.0.5).

Statistical analysis: Data were analyzed using IBM SPSS Statistics (version 29.0.2; IBM Corp., Armonk, NY, USA). The CFU count comparison between samples derived from men and women was performed using an independent samples *t*-test. Pearson’s correlation coefficient was calculated to assess the correlation between age and CFU counts. The level of significance was set at 0.05.

## 5. Conclusions

With this study, we have shown that among skin isolates, staphylococci are the main NT producers. These staphylococcal isolates exhibit different product spectra, specifically the TRY–PEA and TYM–Dopa types. The first type is produced by the SadA enzyme, while the TYM–Dopa type is generated by the TDC enzyme. These AADC enzymes are phylogenetically distinct and differ in size, structure, and the genomic localization of their respective genes. SadA orthologues were found to be more abundant than TDCs among *Staphylococcus* spp., irrespective of their habitat. NTs produced by human microbiota represent a surplus to the endogenous NTs. Interestingly, the TRY and PEA produced by SadA can cross the BBB, while the TYM and Dopa produced by TDC cannot. However, further studies are required to understand the potential contributions of microbiota-derived NTs to overall human physiology and psychology.

## Figures and Tables

**Figure 1 ijms-25-12345-f001:**
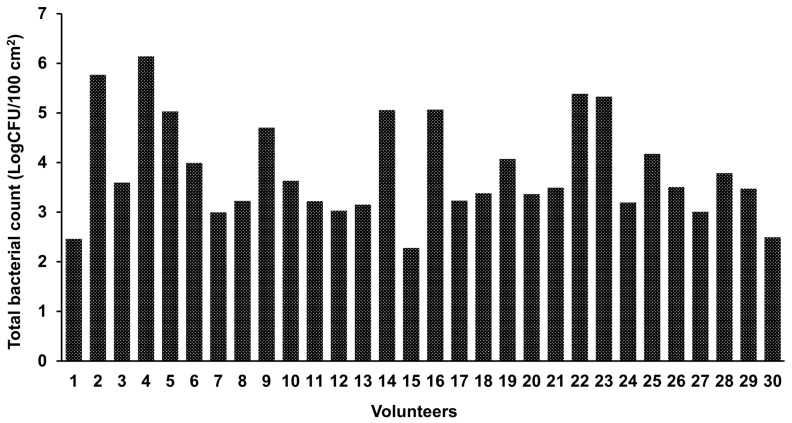
Total bacterial count in the antecubital fossa area from 30 healthy subjects. Numbers represent the volunteer identifiers.

**Figure 2 ijms-25-12345-f002:**
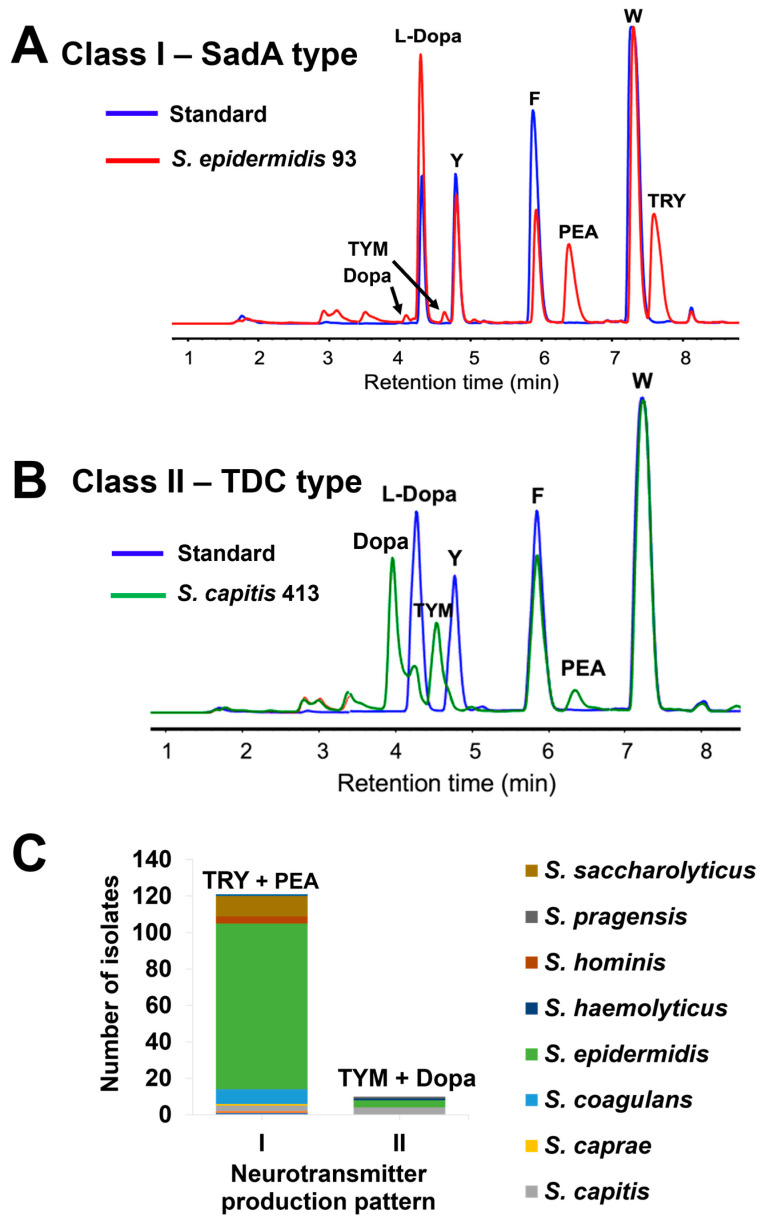
Representative RP-HPLC chromatograms of neurotransmitter production by skin isolates with two different patterns in 5 × PBS buffer/10% glucose and all the substrates (1 mg/mL each): (**A**) *S. epidermidis* 93 produces primarily TRY and PEA, with almost no TYM and Dopa; (**B**) *S. capitis* 413 produces mainly TYM and Dopa, with little PEA and no TRY; (**C**) species distribution of neurotransmitter-producing skin isolates (n = 128) with two production patterns: (**I**) the production of TRY and PEA as main products, with TYM and Dopa at lower levels; (**II**) the production of TYM and Dopa as main products, with PEA at lower levels. Additionally, 5 × PBS buffer/10% glucose supplemented with substrates (1 mg/mL each) was used as the standard. W, tryptophan; F, phenylalanine; Y, tyrosine; TRY, tryptamine; PEA, phenethylamine; TYM, tyramine; Dopa, dopamine.

**Figure 3 ijms-25-12345-f003:**
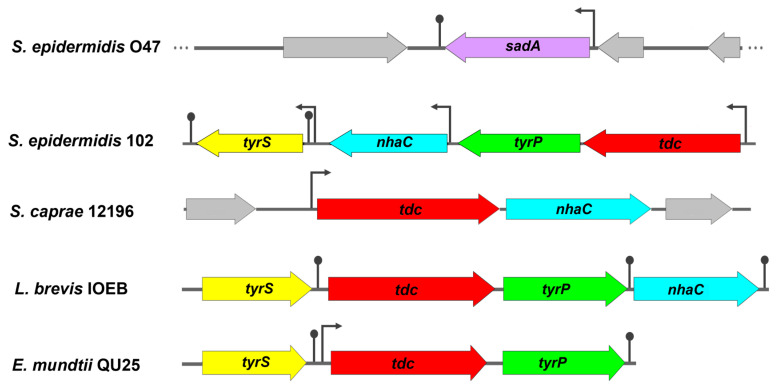
The genetic organization of the tyrosine decarboxylase operons in *S. epidermidis* 102 (this study), *S. caprae* NCTC 12196 (GenBank accession no. AF446085.5), *Levilactobacillus brevis* IOEB (GenBank accession no. AF446085.5), and *E. mundtii* QU 25 (GenBank accession no. AP013036.1) in comparison with *sadA* gene location in *S. epidermidis* O47 (GenBank accession no. CP040883.1). Putative promoters are represented by broken arrows and transcription terminator regions by lollipops. The surrounding genes are shown in gray. *sadA*, staphylococcal aromatic amino acid decarboxylase gene (lilac); ***tdc***, tyrosine decarboxylase gene (red); *tyrP*, tyrosine–tyramine permease gene (green); *nhaC*, Na^+^/H^+^ antiporter gene (blue); *tyrS*, tyrosyl-tRNA synthetase gene (yellow).

**Figure 4 ijms-25-12345-f004:**
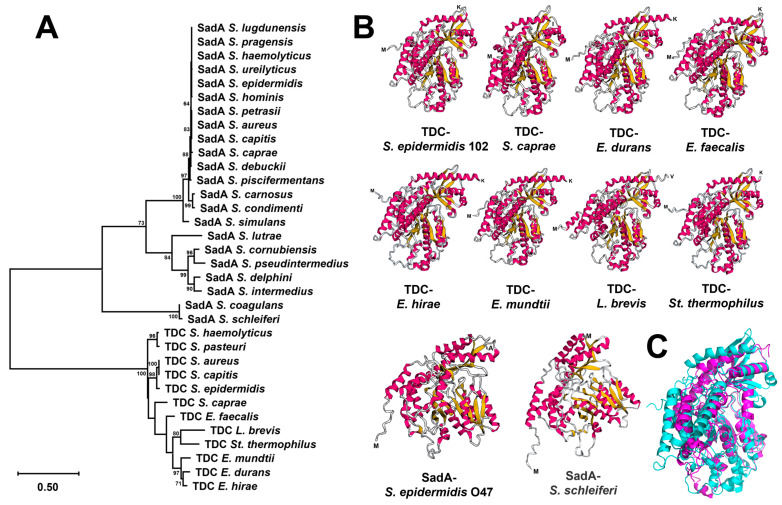
The comparison of SadA and TDC homologs of different bacterial species. (**A**) The maximum likelihood dendrogram of TDC enzymes from *S. epidermidis* 102 (this study), *S. aureus* (GenBank accession no. HDA2154217.1), *S. capitis* (GenBank accession no. WVQ24098.1), *S. caprae* NCTC 12196 (GenBank accession no. AF446085.5), *S. haemolyticus* (GenBank accession no. MBW5904582.1), *S. pasteuri* (NCBI Reference Sequence WP_154840952.1), *Enterococcus durans* IPLA 655 (GenBank accession no. CAF33980), *E. faecalis* JH2-2 (GenBank accession no. AAM46082), *E. hirae* 508 (GenBank accession no. AAQ73505.1), *E. mundtii* QU 25 (GenBank accession no. BAO05941.1), *Levilactobacillus brevis* IOEB (GenBank accession no. AAN77279), and *Streptococcus thermophilus* 1TT45 (GenBank accession no. CBW46640.1) strains and of SadA enzymes from *S. aureus* (GenBank accession no. WP_032605060.1), *S. capitis* Sc1516941 (GenBank accession no. WP_032605060.1), *S. caprae* N99 (GenBank accession no. MBU5272604.1), *S. carnosus* NCTC 13825 (GenBank accession no. SUL89052.1), *S. coagulans* DSM 6628 (GenBank accession no. PNZ10338.1), *S. condimenti* NCTC 13827 (GenBank accession no. VEG62997.1), *S. cornubiensis* NW1 (GenBank accession no. WP_086428678.1), *S. debuckii* SDB 2975 (GenBank accession no. AYU54119.1), *S. delphini* NCTC 12225 (GenBank accession no. VED61613.1), *S. epidermidis* O47 (GenBank accession no. QKN61770.1), *S. haemolyticus* acroh (GenBank accession no. MCI2935196.1), *S. hominis* SDD3 (GenBank accession no. TRM05964.1), *S. intermedius* NCTC 11048 (GenBank accession no. SUM45602.1), *S. lugdunensis* (GenBank accession no. MDU3709123.1), *S. lutrae* ATCC 700373 (GenBank accession no. ARJ50788.1), *S. petrasii* P5404 (GenBank accession no. TGE14952.1), *S. piscifermentans* NCTC 13836 (GenBank accession no. SNU95601.1), *S. pragensis* CCM 8529 (GenBank accession no. GGG93612.1), *S. pseudintermedius* ED99 (GenBank accession no. WP_014612792.1), *S. schleiferi* NCTC 12218 (GenBank accession no. CAD7359406.1), *S. simulans* G12 (GenBank accession no. MCD8915340.1), and *S. ureilyticus* 498-2 (GenBank accession no. MEX6179459.1). The tree was generated using the MEGA_11.0.13 package with 1000 bootstrap replicates. Bootstrap values greater than 60% are listed as percentages at the nodes, and the scale bar represents genetic distance; (**B**) the 3D structure of TDC enzymes from *S. epidermidis* 102 (this study), *S. caprae* NCTC 12196 (GenBank accession no. AF446085.5), *Enterococcus durans* IPLA 655 (GenBank accession no. CAF33980), *E. faecalis* JH2-2 (GenBank accession no. AAM46082), *E. hirae* 508 (GenBank accession no. AAQ73505.1), *E. mundtii* QU 25 (GenBank accession no. BAO05941.1), *Levilactobacillus brevis* IOEB (GenBank accession no. AAN77279), and *Streptococcus thermophilus* 1TT45 (GenBank accession no. CBW46640.1) strains in comparison with SadA enzymes from *S. epidermidis* O47 (GenBank accession no. QKN61770.1), *S. pseudintermedius* ED99 (GenBank accession no. WP_014612792.1), and *S. schleiferi* NCTC 12218 (GenBank accession no. CAD7359406.1) using AlphaFold 2.0. The TDC enzyme from *Lactiplantibacillus plantarum* IR BL0076 (GenBank accession no. AFQ52525.1) was not included in these analyses due to its 100% identity with that of *L. brevis* IOEB; (**C**) the alignment of 3D structures of the TDC enzyme from *S. epidermidis* 102 (cyan) and the SadA enzyme from *S. epidermidis* O47 (GenBank accession no. QKN61770.1; magenta) using PyMOL software (version 3.0.5). The enzymes showed low structural similarity, with a root mean square deviation (RMSD) value of 4.410 Å.

**Figure 5 ijms-25-12345-f005:**
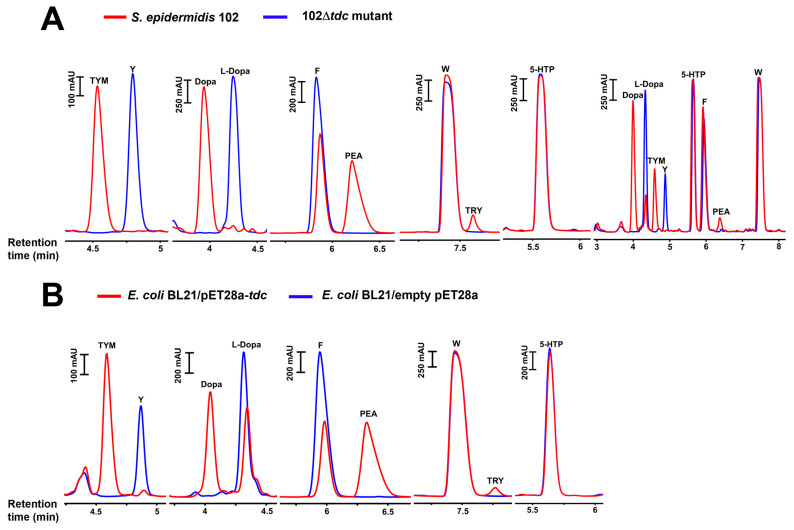
RP-HPLC chromatograms of neurotransmitter production by (**A**) *S. epidermidis* 102 and its *tdc* deletion mutant (102Δ*tdc*) after overnight incubation in 5xPBS buffer/10% glucose containing individual or a combination of substrates (1 mg/mL each); (**B**) *E. coli* BL21 transformants harboring the *tdc* gene from *S. epidermidis* 102 in the vector pET28a after overnight incubation in 5 × PBS buffer/10% glucose containing individual substrates (1 mg/mL each) in comparison with transformants harboring an empty plasmid. 5-HTP, 5-hydroxytryptophan; W, tryptophan; F, phenylalanine; Y, tyrosine; TRY, tryptamine; PEA, phenethylamine; Dopa, dopamine; TYM, tyramine.

**Figure 6 ijms-25-12345-f006:**
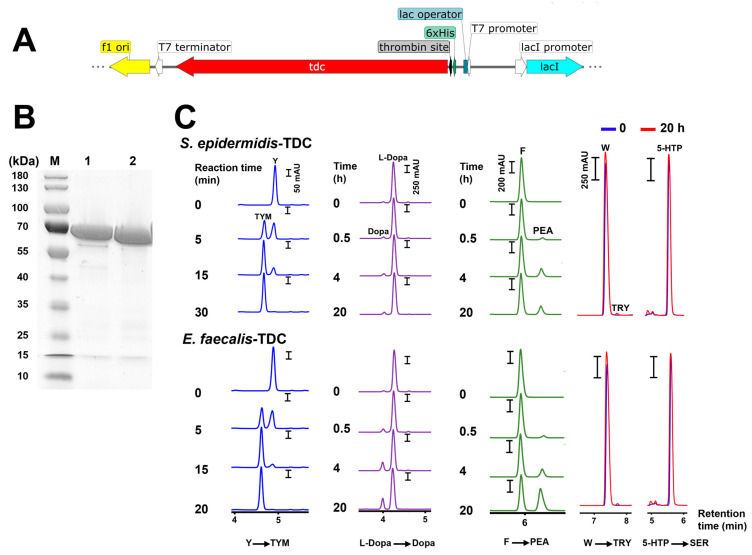
Product patterns with purified TDC enzymes from a *Staphylococcus* and *Enterococcus* strain. (**A**) The section of the pET28a vector with the *tdc* gene from *S. epidermidis* 102 or *Enterococcus faecalis* ATCC 19433, expressed with an N-terminal His tag in *E. coli* BL21; (**B**) the SDS–PAGE analysis of purified TDC enzymes from the *E. coli* clones: M, molecular weight marker; Lane 1, *E. faecalis* ATCC 19433 (GenBank accession no. WP_002355450.1); Lane 2, *S*. *epidermidis* 102; (**C**) RP-HPLC chromatograms of tyramine (TYM), dopamine (Dopa), phenethylamine (PEA), tryptamine (TRY), and serotonin (SER) production by N-terminal His-tagged TDCs of *S*. *epidermidis* 102 and *E. faecalis* ATCC 19433 cloned into pET28a and purified from *E. coli* BL21 transformants in 50 mM sodium phosphate/300 mM NaCl buffer (pH 6.8) containing 150 nM enzyme, 40 µM pyridoxal phosphate (PLP), and 2 mM substrate at 37 °C. The conversions of Y to TYM, L-Dopa to Dopa, and F to PEA are shown in blue, purple, and green, respectively. 5-HTP, 5-hydroxytryptophan; W, tryptophan; F, phenylalanine; Y, tyrosine.

**Table 1 ijms-25-12345-t001:** Spectrum of Gram (+) bacteria isolated from the antecubital fossa skin of 30 healthy subjects.

*Bacillus*	*Staphylococcus*	*Corynebacterium*
*B. agri*	*S. capitis*	*C. bouchesdurhonense*
*B. albus*	*S. caprae*	*C. curieae*
*B. altitudinis*	*S. coagulans*	*C. gottingense*
*B. atrophaeus*	*S. epidermidis*	*C. kefirresidentii*
*B. canaveralius*	*S. haemolyticus*	*C. meitnerae*
*B. cereus*	*S. hominis*	*C. mucifaciens*
*B. haynesii*	*S. petrasil*	*C. parakroppenstedtii*
*B. licheniformis*	*S. pragensis*	*C. pilbarense*
*B. mobilis*	*S. saccharolyticus*	*C. tuberculostearicum*
*B. paramycoides*	*S. saprophyticus*	*C. ureicelerivorans*
*B. siamensis*		
*B. tequilensis*	*Cutibacterium*	*Peribacillus*
*B. tyonensis*	*Cut. acnes*	*P. butanolivorans*
*B. velezensis*	*Cut. avidum*	*P. frigoritolerans*
*B. wiedmannii*		*P. simplex*
*Kocuria*	*Nialia*	*Roseomonas*
*K. arsenatis*	*N. circulans*	*R. mucosa*
*Micrococcus*	*Paeniibacillus*	*Streptococcus*
*M. endophyticus*	*P. etheri*	*St. anginosus*
*M. luteus*		*St. thermophilus*

**Table 2 ijms-25-12345-t002:** Spectrum of trace amine production in 128 skin isolates from healthy subjects after overnight incubation in TSB.

Production Pattern	Species	No. of Isolates
TRY + PEA + TYM	*S. capitis*	1
	*S. coagulans*	8
	*S. epidermidis*	75
	*S. hominis*	4
	*S. saccharolyticus*	11
	Total	99
TRY + PEA	*S. capitis*	2
	*S. caprae*	1
	*S. epidermidis*	16
	Total	19
PEA + TYM	*S. epidermidis*	3
	*S. haemolyticus*	1
	*S. pragensis*	1
	Total	5
TYM	*S. capitis*	4
	*S. epidermidis*	1
	Total	5

TRY, tryptamine; PEA, phenylethylamine; TYM, tyramine.

**Table 3 ijms-25-12345-t003:** Distribution of SadA and TDC homologues in available sequence data from *Staphylococcus* spp.

Enzyme	Species																								
	*S. aureus*	*S. capitis*	*S. caprae*	*S. carnosus*	*S. coagulans*	*S. condimenti*	*S. cornubiensis*	*S. debuckii*	*S. delphini*	*S. epidermidis*	*S. haemolyticus*	*S. hominis*	*S. intermedius*	*S. lugdunensis*	*S. lutrae*	*S. pasteuri*	*S. petrasii*	*S. piscifermentans*	*S. pragensis*	*S. pseudintermedius*	*S. schleiferi*	*S. simulans*	*S. ureilyticus*	*Staphylococcus* sp.*	Total
SadA	676	1	3	14	63	9	1	1	42	798	17	14	5	2	1	ND	1	1	1	4188	55	2	3	24	5922
TDC	1	21	30	ND	ND	ND	ND	ND	ND	125	14	ND	ND	ND	ND	1	ND	ND	ND	ND	ND	ND	ND	6	198

ND, not detected; SadA, staphylococcal aromatic amino acid decarboxylase; TDC, tyrosine decarboxylase. * No specified species.

**Table 4 ijms-25-12345-t004:** BLASTp sequence homology analyses of *tdc* operons from *Staphylococcus epidermidis* 102 and *Levilactobacillus brevis* IOEB.

*tdc* Operon				Blastp		
*S. epidermidis* 102		*L. brevis* IOEB				
Protein	Length (aa)	GenBank Accession No.	Length (aa)	Coverage (%)	E Value	Identity (%)
Tyrosyl-tRNA synthetase (TyrS)	417	AAQ83557.1	418	99	0.0	69
Tyrosine decarboxylase (TDC)	616	AAN77279.2	635	95	0.0	71
Tyrosine–tyramine permease (TyrP)	479	AAQ83558.1	473	99	0.0	66
Na^+^/H^+^ antiporter (NhaC)	461	AAQ83559.1	476	96	2 × 10^−142^	55

## Data Availability

The whole genome sequence of *S. epidermidis* 102 was deposited in GenBank under the accession number JBIENV000000000.

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
