# Peer review of "Diversity of Neurotransmitter-Producing Human Skin Commensals"

_ijms, 2024, doi:10.3390/ijms252212345_

Round 1
Reviewer 1 Report
Comments and Suggestions for Authors
Dear authors
The manuscript „Diversity of Neurotransmitter Producing Human Skin Commensals”The secretion, complex function and physiological role of neurotransmitters are incompletely elucidated, but new discoveries are made regarding the intestinal tract, vascular endothelium and skin. This study is a very interesting study, touching on the key points of the study of neurotransmitters in the skin. Thus:
The abstract is properly structured, contains relevant information that correctly describes the study. No improvement required.
The introduction describes from the correctly collected literature, both biogenic amines and neurotransmitters. I was delighted by the use of the term monoamines, noting that the authors have deepened the understanding of what amines represent in the role of NT secretion. The description in the introduction of the most important enzymes involved in NT secretion, involving the bacterial secretion at the skin level, makes the introduction complete. The object of the study is clear.
The results are presented coherently. The description of Bacterian colonies, and the descriptive statistics applied, is correctly presented, underlined by clear, easy-to-follow figures and tables. the description of the identification process of secretory NT bacteria is clear, easy to follow from table 2 and figure 2.
The identification of the enzymes involved in NT secretion at the skin level is for me the most relevant part, correctly presented. Please add a general title to figure 4, before the description of the subcategories.
The enzymes responsible for NT secretion are complexly presented in both figure 5 and figure 6, from which the reader can receive concrete and complete information.
Materials and methods include the description of each method, complete with all information to make the study reproducible.
The correctly conducted discussion achieves every result obtained, to which only a small part of limitations can be added.
The conclusions are clear, they include the conclusions of the correctly extracted results and the perspectives regarding future studies.
Congratulations on the work!
Reviewer 2 Report
Comments and Suggestions for Authors
In the manuscript entitled “DIVERSITY OF NEUROTRANSMITTER PRODUCING HUMAN SKIN COMMENSALS”, the authors present the results of their research on skin neurotransmitters produced by human skin commensals. Thus, considering that human microbiota can excrete trace amines, dopamine, and serotonin, the authors mention that these neurotransmitters can either affect classical neurotransmitter signaling or directly trigger trace amine-associated receptors (TAARs), with still unclear consequences for host physiology. Thus, compared to gut microbiota, less information is available on the role of skin microbiota in neurotransmitter production. To explore this, in this research 1909 skin isolates, mainly from the genera Staphylococcus, Bacillus, and Corynebacterium, were tested for neurotransmitter production. So, only 6.7% of the isolates were capable of producing NTs, all of which belonged to the Staphylococcus genus. Based on substrate specificity, the authors identified two distinct profiles among the NT producers. One group primarily produced tryptamine (TRY) and phenylethylamine (PEA), while the other mainly produced tyramine (TYM) and dopa-mine (Dopa).+ Also, the authors mention that the genomic localization of the respective genes also varies: tdc genes are typically found in small, conserved gene clusters, while sadA genes are not. Heterologous expression of sadA and tdc in Escherichia coli yielded the same product spectrum as the parent strains.
Thus, this manuscript could be a valuable article which presents this interesting topic and involves meaningful author results. However, there are some suggestions:
-
What is currently known about neurotransmitters in skin (basic/key knowledge)? Which skin diseases are mentioned as dermatoses which involve participation of neurotransmitters? You may mention at least some examples of these dermatoses. In medical literature, it is possible to see their different roles, as well as interactions with other immune and endocrine factors. However, it is not mentioned in the manuscript, although it is important for this topic and interpretation of the results.
-
Materials and Methods: What about data that the research was approved by the local Ethical Committee, which follows principles of good clinical practice?
-
Figure 1. I suggest to add data that these numbers are the numbers of each examinee – to be more clear to the reader.
-
Discussion: What is the practical meaning of this research for clinical practice? What is important in this research as a basis for future research?
-
I suggest adding limitations and advantages to this research.
-
The last part of the Conclusion in mores suitable for Discussion section:
„Interestingly, the TRY and PEA produced by SadA can cross the BBB, while the
TYM and Dopa produced by TDC cannot. This means that the TAs produced by the majority of Staphylococcus species can reach the CNS by being taken up through the skin. Since S. aureus infections are known to cause a variety of neurological complications, most of which affect the CNS [65], the question is to what extent TRY and PEA contribute to these effects. However, these processes are still poorly understood and present challenges“
However, I can comment on the text as a clinician and I think that it would be useful to check the obtained data by an expert from molecular biology and similar fields.
Reviewer 3 Report
Comments and Suggestions for Authors
Congratulations on the written manuscript.
1. The main question of the research is to find out the ability of human skin microbiota to produce neurotransmitters, as well as to identify bacteria. The authors investigate the specificity of various enzymes responsible for the synthesis of certain neurotransmitters.
2. Do you consider the topic original or relevant to the field? Does it address a specific gap in the field?
Yes, this topic is original and very relevant. The study helps to fill this gap by highlighting the possible influence of skin bacteria on the neurosignaling pathways of the host.
3. What does it add to the subject area compared with other published material?
This research is an important addition to the field because it: Identifies neurotransmitter-producing strains of skin bacteria, primarily Staphylococcus. Determines different profiles of neurotransmitter production, depending on enzymatic specificity. Shows how genetic variations, such as the length and location of the sadA and tdc genes, affect the ability of bacterial strains to synthesize certain neurotransmitters.
4. What specific improvements should the authors consider regarding the methodology?
To increase the number of bacterial isolates and/or to expand the generic diversity, to carry out whole-genome sequencing of the most interesting strains. This is a suggestion, not a comment.
5. Are the conclusions consistent with the evidence and arguments presented?
In general, the conclusions are justified and correspond to the obtained data. However, conclusions regarding possible effects on human physiology are somewhat speculative (lines 418-419). This kind of assumption should be discussed in the discussion section, and not in the conclusions section, especially since the interaction with the central/peripheral nervous systems was not the subject of the study, it is a hypothesis. Therefore, it is useful to include references to previous studies revealing such a relationship or suggesting a mechanism of action in the discussion section.
6. Are the references appropriate?
Yes.
7. Any additional comments on the tables and figures?
No.
8. Line 90-92. This is about the results, not the introduction, maybe you should move those sentences to the next section.
9. It is worth adding a section or a paragraph about study limitations.
